# Differences in Screen Addiction in the Past 15 Years

**DOI:** 10.3390/ijerph21010001

**Published:** 2023-12-19

**Authors:** Núria Aragay, Vicenç Vallès, Irene Ramos-Grille, Gemma Garrido, Enric Gamundi Grimalt, Elena Miranda Ruiz, Esther Jovell-Fernández

**Affiliations:** 1Behavioral Addictions Unit, Area of Mental Health Care, Consorci Sanitari de Terrassa, 08227 Terrassa, Spain; vvalles@cst.cat (V.V.); iramos@cst.cat (I.R.-G.); 2Faculty of Medicine and Health Sciences, Universidad Internacional de Catalunya, 08017 Barcelona, Spain; ggarrido@cst.cat (G.G.); ejovell@cst.cat (E.J.-F.); 3Area of Mental Health Care, Consorci Sanitari de Terrassa, 08227 Terrassa, Spain; emiranda@cst.cat; 4Community Rehabilitation Service, Area of Mental Health Care, Consorci Sanitari de Terrassa, 08227 Terrassa, Spain; 5Hematological Cytology, Pathological Anatomy Service, Hospital Universitario de Bellvitge, 08907 Barcelona, Spain; egamundi@bellvitgehospital.cat; 6Department of Epidemiology, Consorci Sanitari de Terrassa, 08227 Terrassa, Spain

**Keywords:** video games, internet, social networks, addiction

## Abstract

The use of information and communication technologies (ICTs) has become widespread in recent years, leading to an increase in addiction cases. A total of 118 patients who attended the Behavioral Addictions Unit of Consorci Sanitari de Terrassa (Barcelona, Spain) between October 2005 and December 2021 were included in the study. The sample was divided into three groups according to the time period in which they started treatment: between 2005 and 2010 (before the rise in new technologies, named the pre-ICT period), between 2011 and 2019 (a time of major ICT development, named the ICT period) and between 2020 and 2021 (with massive use of the internet due to effects of the pandemic, named the COVID-19 period). We found an increase in the incidence of screen addiction cases related to the development of technology throughout the study years, and this increase was accentuated during the COVID-19 period. This increase was not equal for all types of content consumed via the internet, with patients with video game addiction increasing to a greater extent than patients with internet/social network addiction. In addition, patients with video game addiction were younger and had started gaming at a younger age than those with internet/social network addiction. These results contribute to a better understanding of the phenomenon of technology addiction and to the design of appropriate treatment protocols and preventive programs.

## 1. Introduction

Accessing the internet, playing video games, or participating in social networks are very common leisure activities in our society. The internet is a useful and entertaining tool that eases access to information and allows multiple possibilities for social interaction. The general population lives permanently connected, but it is young people who are the most connected as they regularly socialize through mobile and computer screens. According to the Drug Use Survey in Secondary Education in Spain (ESTUDES) and Survey Program on Alcohol and Other Drugs in Spain (EDADES) in 2021, the prevalence of internet use in Spain among individuals aged 15 to 44 is 98% [1]. Thus, young people are at higher risk of developing potential problems related to screen use.

Information and communication technologies (ICTs) have evolved considerably over recent years, through a rich variety of multimedia content, interactive platforms, and virtual experiences. All these kinds of content provide a personalized and immersive digital environment designed to elicit pleasure, reward, and emotional engagement. These attributes contribute to a substantial hedonic component designed to generate wellbeing and help us to escape from reality.

The inception of multiplayer video games paved the way for the emergence of massively multiplayer online role-playing games (MMORPGs) in 2002, enabling thousands of players to coexist in virtual worlds simultaneously [2]. The first social networks appeared between 2005 and 2010 (LinkedIn, Facebook, YouTube, WhatsApp and Instagram), but this was a period when the internet was mainly a tool for information storage and communication. Subsequently, between 2011 and 2019, there was a period of change and evolution in the content offered through the internet, with a boom in ICTs and especially in the use of MMORPGs and social networks. In addition to this change in content, the devices with which we access the internet have evolved to the latest generation of tablets and mobile phones, which allow us to swiftly access a wide range of content. Finally, the COVID-19 pandemic inevitably changed the use of ICTs, increasing internet use with a prevalence rate of 10.6% for internet addiction, 5.3% for gaming addiction, and 15.1% for social media addiction [3]. However, the clinical impact of this global situation in terms of behavioral addiction and characteristics of addiction is unknown [4]. 

The changes experienced by digital technology, together with its widespread use, have led to concerns about its excessive use. The omnipresence of digital devices and constant connectivity can lead to heightened levels of stress, anxiety, and even addiction in both young people and adults. However, the different studies that have attempted to analyze the prevalence of the problems associated with its use are heterogeneous, both in terms of the samples used (students, user population, general population, etc.) and the behavior analyzed (addiction to internet, video games, social networks, or mobile phones) [5,6,7]. This heterogeneity explains the large disparity in addiction prevalence rates, with, for example, a reported prevalence of social network addiction ranging from 5% to 25% [8,9,10,11]. 

Currently, there is a lack of consensus when considering screen or internet addiction as an addictive disorder [2,6], although some authors consider that digital access can lead to adverse consequences like those produced by substance addictions [5,12]. Moreover, there is controversy about whether the addiction is to the device (mobile phone, computer, tablet) or to the content consumed (video games, social networks, etc.) [13]. Some authors suggest that there is a “pathological use of the internet”, which is considered a behavioral addiction defined as the loss of control over one’s technology use and connection-seeking behavior that generates negative consequences and the impossibility of curbing the impulse or desire to do so [14,15]. However, these studies include a variety of uses (video games, social networks, different devices, etc.) within the same concept of excessive internet use. On the other hand, other authors question its existence as a nosological entity and consider that the internet is only the channel through which we access to this content [2,6]. However, there is more consensus about the adverse consequences of video games. In fact, the Diagnostic and Statistical Manual of Mental Disorders (DSM-5) of the American Psychiatric Association (APA) already considered video game disorder as “Internet Gaming Disorder” in the 2013 edition; despite this consideration, it was taken into account that this disorder required further studies to define its classification, and it was specifically classified in “Section III: Emerging measures and models—Conditions for further study” [16]. Some years later, the WHO included “Video Game Addiction” in the latest revision of the International Classification of Diseases (ICD-11) [17]. However, no other types of related addiction such as isolated internet addiction or social network addiction were considered in these manuals [15,16,17].

Current evidence does not provide a basic theory of this type of disorder nor validated and standardized definitions and evaluation tools, which leads to a limited understanding of the nature and severity of associated screen-use disorders. Furthermore, most of the research on this subject has focused on the general population rather than incorporating patients [6]. Herein, we present a study focusing on epidemiological and clinical characteristics of a series of patients with different screen addictions, aiming to assess the evolution and patterns of screen addiction in the past 15 years.

## 2. Materials and Methods

### 2.1. Study Design

This is an observational study on a prospective cohort including all consecutive patients who attended to the Behavioral Addictions Unit of Consorci Sanitari de Terrassa (Barcelona, Spain), which provides psychological care for a population of 1 million inhabitants, from October 2005 to December 2021. All patients with problems of abuse or addiction to any internet content (social networks, video games, etc.) were included. Exclusion criteria were patients diagnosed with gambling disorder according to DSM-IV-TR or DSM-5, as appropriate, and patients diagnosed with other behavioral addictions, such as shopping or sex addiction. Subjects gave their consent for participation in the study, and data were collected in accordance with the Spanish Data Protection Act (Organic Law 3/2018, of 5 December, on Personal Data Protection). The study was approved by the Clinical Research Ethics Committee of Consorci Sanitari de Terrassa (approval number: 05-22-107-110).

The main objective of our study is to analyze the temporal evolution of patient profiles among those treated for screen-use problems since the appearance of these new technologies, including the period of the COVID-19 pandemic. As a secondary objective, we aim to analyze the differential clinical characteristics of patients according to their main internet addiction.

### 2.2. Variables and Outcomes

All patients were treated and followed up by a team of psychologists, supervised by a senior clinical psychologist with more than 15 years’ experience in the diagnosis and treatment of behavioral addictions.

Information was recorded on socio-demographic variables: age, sex, cohabitation, employment status. Variables related to addictive behavior were also recorded: type of addiction, age of onset of the behavior, age of onset of addiction, frequency of the behavior, and previous attempts to stop the behavior. Finally, we recorded clinical variables assessed according to the DSM-5.

To analyze the evolution of the cases, the sample was divided into three groups according to the time period in which they started treatment: those attending between 2005 and 2010 (when there had not yet been a rise in new technologies, named the pre-ICT period), those attending between 2011 and 2019 (a period of major ICT development, named the ICT period) and those attending between 2020 and 2021 (massive use of the internet due to the effects of the pandemic, named the COVID-19 period).

To analyze clinical characteristics, patients were divided into two groups: those patients with internet and/or social network addictions versus those patients with video game addiction.

### 2.3. Statistical Analysis

A descriptive analysis was performed presenting the absolute and relative frequency for qualitative variables and the mean with standard deviation (±SD) for quantitative variables. The prevalence of different addictions was compared through the three mentioned periods (pre-ICT, ICT, and COVID-19 periods) and baseline clinical characteristics were compared between patients with internet and/or social network addictions versus those with video game addiction.

Categorical variables were compared with the Chi-square test or the Fisher’s exact test, whereas the Student’s t-test or analysis of variance were used to compare continuous variables. The Kolmogorov–Smirnov test was performed for continuous variables to assess normality. Non-parametric variables were compared using the Mann–Whitney U test. A value of *p* < 0.05 was considered statistically significant. Analyses were performed with IBM SPSS Statistics, version 25.0 for the PC (IBM Corp., Armonk, NY, USA).

## 3. Results

### 3.1. Global Sample

During the study period, an average of 100 new patients per year attended our referral unit, resulting in a total of 1262 patients with different types of behavioral addiction. Of this total, 118 (9.4%) patients with screen addiction were included. Mean age at the time of the first visit was 23 ±10 and 108 (91.5%) were male. Mean age of the onset of screen-using behavior was 17 (SD: 10). At the time of starting treatment, 79.7% (*n* = 94) were living with their parents and 60.2% (*n* = 71) were studying. In terms of addictive behavior, 23 (19.5%) patients attended for internet and/or social network addiction and 95 (80.5%) patients attended for video game addiction. In 95.8% (*n* = 113) of the cases, the frequency of the behavior was daily, 74.6% (*n* = 88) had been presenting addiction problems for more than a year, and 74.6% (*n* = 88) had never made any previous attempt to stop the addictive behavior (Table 1).

### 3.2. Temporal Evolution of the Screen Addiction Profile

Since 2005, the number of new patients attending our unit for screen addiction has increased. During the pre-ICT period (between 2005 to 2010), a total of 19 (16.1%) patients began treatment for these disorders, while in the ICT period (between 2011 to 2019), there was a substantial rise in the number of patients, reaching 68 (57.6%) new patients. Notably, during the two-year COVID-19 period, the unit experienced a surge of 31 (26.3%) new patients, highlighting the impact of the COVID-19 pandemic on the incidence of this disorder. The number of new patients year by year is shown in Figure 1. Regarding the annual incidence of the different types of screen addiction, while the number of patients/year with internet and/or social network problems remained similar in the pre-ICT, ICT, and COVID-19 periods (5 patients/6 years = 0.8, 13 patients/9 years = 1.4, and 5 patients/2 years = 2.5, respectively), the number of patients/year with video game addiction increased gradually (14 patients/6 years = 2.3, 55 patients/9 years = 6.1, 26 patients/2 years = 13, respectively).

When we compared patients through the three mentioned periods (pre-ICT, ICT, and COVID-19 periods), we found that patients had a lower mean age of onset of screen-using behavior during the COVID-19 period (between 2020 to 2021) than in the ICT period (2011–2019), 11 ± 5 vs. 19 ± 12 years, respectively (*p* = 0.041). In addition, the frequency of daily screen use was higher during the COVID-19 period compared with the pre-ICT and the ICT periods (100% vs. 84.2% vs. 97.1%, respectively; *p* = 0.019) (Table 1).

### 3.3. Differential Clinical Characteristics of Patients According to Their Primary Screen Addiction

When comparing patients with a primary addiction to video games (*n* = 95) with those with internet and/or social network addiction (*n* = 23), the former had a lower mean age of onset of addictive behavior (15 ± 8 vs. 26 ± 14; *p* < 0.001) and a lower mean age at the first visit (20 ± 8 vs. 31 ± 13; *p* < 0.001) (Table 2, Figure 2). In addition, significant differences were found in cohabitation and in employment status. Patients with video game addiction lived more frequently with their parents or family of origin than those with internet and/or social network addiction (97.9% vs. 65.2%, respectively; *p* = 0.000). Likewise, patients with video game addiction were more likely than the other group to be studying as opposed to working (68.4% vs. 26.1%, respectively; *p* = 0.000) (Table 2).

## 4. Discussion

Our study reveals a notable surge in the prevalence of screen addiction, particularly in the context of video games. This increase was further amplified during the COVID-19 pandemic period. Previous studies have analyzed samples of internet users, such as students or the general population, but not of subjects requiring treatment for screen addiction [1,5]. These studies helped to highlight the rise and growth in the use of technology and allowed the identification of characteristics and factors related to its overuse. However, the lack of studies with clinical samples including patients and not focusing only on the general population, the disparity of instruments used for the diagnosis of addiction, and the heterogeneity of the samples analyzed contribute to a lack of consensus in terms of diagnostic criteria and classification of screen-related disorders. This scenario makes difficult the early and accurate diagnosis of these disorders, as well as the implementation of treatment strategies and preventive campaigns. In this context, our study analyzes patients who have required specialized psychological treatment due to abusive use or addiction to internet, social networks, or video games, that is, from the perspective of the disorder.

The progressive increase in the number of cases that we have observed may be related to the development of technology, both in terms of devices and the content offered through the internet [2]. In our study, the highest number of cases was detected during the years 2010–2019 (ICT period). Kuss et al. also highlighted that the increasing popularity and frequency of internet use has led to an increasing number of users and the potential negative consequences of overuse [5]. In fact, this period saw the emergence and popularization of online platforms such as YouTube, social networks, and MMORPGs. It was a period of technological splendor which led to transformation of devices, the content offered, and its accessibility [18,19]. These new tools and their content can be considered instruments geared towards hedonism, allowing people to escape from their everyday worries. All of this may have contributed to the increase in the number of patients in recent years.

In our study, during the COVID-19 period, the increasing trend of these problems continued. The increase observed during this period was consistent with the results of the study by Alimoradi et al. [3], which analyzed the effects of lockdown during the COVID-19 pandemic on behavioral addictions. The lockdowns resulted in radical changes in the behavioral habits of the world’s population, in many cases forcing them to live via the internet. Work, studies, friendships, and family relationships were all conducted through screens, normalizing such forms of socializing and even making it difficult to return to the former face-to-face normality once the lockdowns were lifted. As behavioral addictions resulted in an important health problem during the COVID-19 pandemic, healthcare providers and government authorities ought to promote campaigns focused on aiding individuals in managing stress in similar potential future scenarios [3,4].

The increase in patients observed in our study was not the same for the different types of content consumed via the internet. While the number of cases seeking treatment for video game addiction increased by six from the pre-ICT period to the COVID-19 period, the increase in the number of patients seeking treatment for internet/social network addiction during these periods was lower, increasing by three. Our results could be attributable to the potential addictive difference between video games and social networks. Video games allow multiple players to enter a virtual world simultaneously and interact with each other. These games have high quality graphics, generating an immersive sensation for the player that contributes to a detachment from reality. In addition, the players have an “avatar” that they can create according to their preferences, which helps to satisfy needs related to the image of one’s “ideal self”, allowing the person concerned to feel integrated and even well considered by the other players. Moreover, as they practice, the players gradually improve, feeling more capable and confident in the game. All these characteristics are largely rewarding and explain the high addictive potential of video games [2]. On the other hand, social networks provide a platform for interactive social engagement, allowing us to express ourselves as “we are” or as “we wish to be”. This characteristic along with the allure of ‘likes’ and the infinite ‘scroll’ endow social networks with a high addictive potential. ‘Likes’ feed our self-esteem, fostering a sense of belonging and acknowledgement from others. Furthermore, the infinite ‘scroll’ offers a constant stream of short videos or photos that can captivate our attention during the whole day, staving off boredom [2,20].

Although both internet/social networks and video games have addictive characteristics, the latter cause greater interference in daily functioning because they require being isolated in a room to hear and talk to other players, and thus interfere with responsibilities and obligations (going to school, extracurricular activities, socializing, etc.). This higher impact on basic daily activities can lead to greater problems of control than internet/social networking problems and may explain the higher increase in video game addiction in our study. In contrast, social networks are less disruptive for daily activities as they can be accessed via mobile phones in different situations. Although excessive use of social media can cause behavioral addiction, it can also manifest as self-esteem or body image disorders, leading to other conditions like anxiety, depression, or obsessive–compulsive disorders as the reason for consultation [21,22,23].

Patients in our study who consulted for video game addiction had initiated the behavior at a younger age and were also younger at the time of seeking treatment than patients who consulted for problems with the internet or social networks. Consequently, these patients live in a familiar environment and are most often at school. Playing is a priority during childhood and this would explain the preference for video games among the youngest, but at that age, due to difficulties in self-regulation and self-control, they are particularly vulnerable to developing addiction problems [24]. At older ages, there is greater interest in communicating and being in contact with peers, which would explain the higher use of social networks in adolescents and young adults [25].

The current study has some limitations. Firstly, it is a single-center study, with a limited sample. However, it offers a view from the perspective of patients treated for addiction to these technologies. Secondly, some more specific clinical or personality variables that could have added information to the patient profile have not been recorded. However, the structured and standardized follow-up guarantees a homogeneous assessment of the analyzed sample, regardless of the type of addiction. Finally, the division of time periods is arbitrary. However, the long overall period of time analyzed allows a temporal assessment of the impact of technology on these addictions, which is an aspect that has been scarcely analyzed previously.

## 5. Conclusions

According to the findings of our study, there has been an increase in the incidence of screen addiction in recent years and this increase was accentuated during the COVID-19 pandemic. Addiction to video games has increased to a greater extent than addiction to the internet or social networks. Patients with video game addiction are younger and tend to start playing at an earlier age than those with internet/social network addiction. These results contribute to a better understanding of screen addictions and to the design of appropriate treatment protocols and preventive programs focusing especially on young people and families. There is a need to implement educational programs that promote a healthy use of technology and prevent screen addictions.

## Figures and Tables

**Figure 1 ijerph-21-00001-f001:**
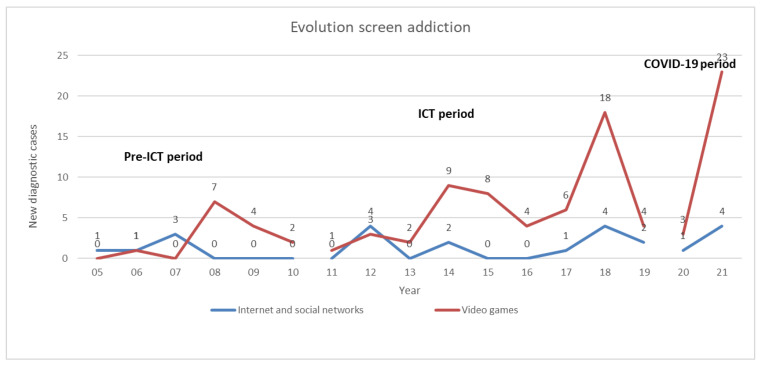
Number of new cases per year managed at the Behavioral Addictions Unit during the study period (2005–2021).

**Figure 2 ijerph-21-00001-f002:**
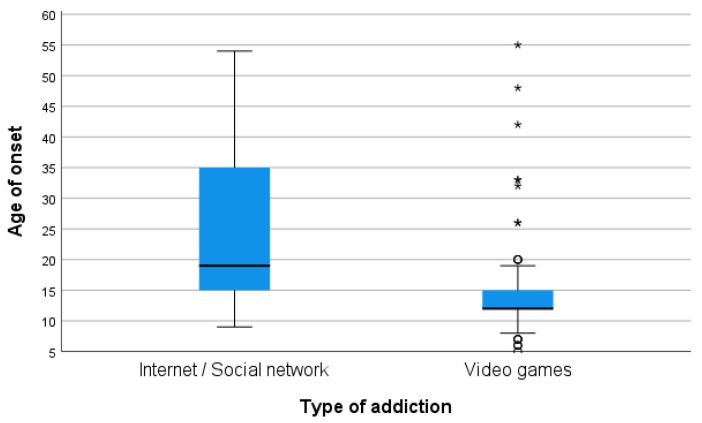
Distribution of age of onset and age at first visit by type of addiction. * severe outliers; ° mild outliers.

**Table 1 ijerph-21-00001-t001:** Patient characteristics according to the time they started treatment.

Characteristics	Total (*n* = 118)	Pre-ICT (*n* = 19)	ICT (*n* = 68)	COVID-19 (*n* = 31)	*p*-Value
Male gender (*n*, *%*)	108 (91.5)	16 (14.8)	65 (60.2)	27 (25)	0.170
Age at onset, mean (SD)	17 (10)	18 (6)	19 (12)	11(5)	0.000 ^1^
Age at first visit, mean (SD)	23 (10)	24 (8)	24 (12)	20 (7)	0.117 ^2^
Main addiction (*n*, %)					0.673
Internet/social network	33 (27.9)	5 (26.3)	13 (19.1)	5 (16.1)	
Video games	95 (80.5)	14 (73.7)	55 (81)	26 (84)	
Cohabitation (*n*, %)					0.208
Parents	94 (79.7)	14 (74)	51 (75)	29 (94)	
Partner and children	9 (7.6)	1 (5.3)	8 (12)	0	
Single	8 (6.8)	3 (16)	3 (4.4)	2 (6.5)	
Partner	6 (5.1)	1 (5.3)	5 (7.4)	0	
Employment status (*n*, %)					0.074
Working	19 (16.1)	4 (21)	9 (13.2)	6 (19.3)	
Studying	71 (60.2)	7 (36.8)	42 (62)	22 (71)	
Neither	28 (23.7)	8 (42.1)	17 (25)	3 (9.7)	
Time out of control (*n*, %)					0.427
<1 year	30 (25.4)	3 (15.8)	17 (25)	10 (32.2)	
>1 year	88 (74.6)	16 (84.2)	51 (75)	21 (67.7)	
Frequency (*n*, %)					0.019
Daily	113 (95.8)	16 (84.2)	66 (97.1)	31 (100)	
Weekly	5 (4.2)	3 (15.7)	2 (2.9)	0	
Previous attempts to quit (*n*, %)					0.220
Yes	21 (17.8)	1 (5.3)	16 (24)	4 (13)	
No	88 (74.6)	14 (73.7)	50 (74)	24 (77.4)	

^1^ effect size: 0.099 (IC_95_: 0.014–0.201). ^2^ effect size: 0.031 (IC_95_: 0.000–0.104).

**Table 2 ijerph-21-00001-t002:** Patient characteristics according to the type of content they were addicted to.

Characteristics	Total (*n* = 118)	Internet/Social Network (*n* = 23)	Video Games (*n* = 95)	*p*-Value
Male gender (*n*, *%*)	108 (91.5)	19 (17.6)	89 (82.4)	0.103
Age at onset, mean (SD)	17 (10)	26 (14)	15 (8)	0.000 ^1^
Age at first visit, mean (SD)	23 (10)	31 (13)	20 (8)	0.000 ^2^
Treatment period (n, %)				0.673
Pre-ICT	19 (16.1)	5 (21.7)	14 (14.7)	
ICT	68 (57.6)	13 (56.5)	55 (57.9)	
COVID-19	31 (26.3)	5 (21.7)	26 (24.4)	
Cohabitation (*n*, %)				0.000
Parents	94 (79.7)	10 (43.5)	84 (93.7)	
Partner and children	9 (7.6)	5 (21.7)	4 (4.2)	
Single	8 (6.8)	4 (17.4)	4 (4.2)	
Partner	6 (5.1)	4 (17.4)	2 (2.1)	
Employment status (*n*, %)				0.000
Working	19 (16.1)	12 (52.2)	7 (7.4)	
Studying	71 (60.2)	6 (26.1)	65 (68.4)	
Neither	28 (23.7)	5 (27.3)	23 (24.2)	
Time out of control (*n*, %)				0.289
<1 year	30 (25.4)	8 (34.8)	22 (23.2)	
>1 year	88 (74.6)	15 (65.2)	73 (76.8)	
Frequency (n, %)				0.582
Daily	113 (95.8)	23 (100)	90 (94.7)	
Weekly	5 (4.2)	0	5 (5.3)	
Previous attempts to quit (*n*, %)				0.769
Yes	21 (17.8)	5 (21.7)	16 (16.8)	
No	88 (74.6)	18 (78.3)	70 (73.7)	
Psychiatric comorbidity (*n*, *%*)	41 (34.7)	5 (21.7)	36 (37.9)	0.133

^1^ effect size: 1.160 (IC_95_: 0.678–1.637). ^2^ effect size: 0.031 (IC_95_: 0.879–1.856).

## Data Availability

Data used for analysis can be supplied from the corresponding author upon request.

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
