# Peer review of "Differences in Screen Addiction in the Past 15 Years"

_ijerph, 2023, doi:10.3390/ijerph21010001_

Round 1
Reviewer 1 Report
Comments and Suggestions for Authors
Please see the attached document with comments and suggestions.

Would benfit from grammatical and structural revision.
Reviewer 2 Report
Comments and Suggestions for Authors
This is an interesting research with a long time scale. Addiction to video games and social media is a worldwide problem and it's good that it's being addressed. Social media addiction is ruining the lives of many families.
The presented article is focused on a very current topic - dependence on ICT. It is written in an understandable way, the results of the investigation are clearly presented. I respect the given form of communication. It is an ICT threat alert. There are other variants of these threats, such as online shopping and dating. But I understand GDPR. I very much appreciate the time span, which documents how dependence on screens gradually changed with the development of ICT, or internet, video games and social networks. These addictions negatively affect the lives of many individuals and entire families, and it is necessary to introduce not only treatment, but also preventive measures, the proposal of which could be supplemented by the authors, e.g. what the school can do in this direction. The increase in these addictions is enormous and a serious problem in many countries. It is good that the authors draw attention to this problem.
The article will certainly be of interest to researchers from other countries as well. I have no comments.
Reviewer 3 Report
Comments and Suggestions for Authors
This manuscript ‘Differences in screen addiction in the last 15 years’ is dedicated an analysis the phenomenon of technology addiction.
The authors the focus is in the incidence of screen addiction cases related to the development of technology throughout the study years (between October 2005 and December 2021), and this increase was accentuated during the COVID-19 pandemic.
The relevance of the work lies in the fact that authors proved that the increase of screen addiction cases was not equal for all types of content consumed via the internet, with video game addiction increasing to a greater extent than internet/social network addiction. In addition, patients with video game addiction were younger and had started gaming at a younger age than those with internet/social media addiction.
I have found the manuscript very interesting and gap filling, but some improvements should be done for a better comprehensive reading.
Comments and Suggestions for Authors:
• Generally the conclusions consistent with the evidence and arguments presented.
• Unfortunately, the authors do not consider recent studies on this topic. For theoretical framework and bibliography additional current references should be included to new research 2022-2023.
• The authors should describe the tools in more detail
• The research topic seems to be very relevant, but the conclusions given in the article would be more convincing if the authors described their practical application or plans for practical application.
• The process of discussing the results can be extended by applying the results and extrapolating them to other similar studies.
• Please describe in detail how your study fits for aims and scope of International Journal of Environmental Research and Public Health.
• Please improve the structure of the article in accordance with the Instructions for Authors. Please check the presence of all structural sections of the article, correct the structure of the article in accordance with the recommendations.
Round 2
Reviewer 1 Report
Comments and Suggestions for Authors
I think the authors have made an effort to improve the document and their effort is appreciated. I believe the tables and figures can use some work/updating but aside from that I am content with the updates.
Comments on the Quality of English LanguageIt can use more English language editing, but it´s improved from the original submission.
Author Response
We appreciate the reviewer’ comments and suggestions and believe that the manuscript has improved by the changes made accordingly. The english language of the manuscript has also been revised, and the modifications made after the language review are highlighted in green.